# The Clinical Need for New Diagnostics in the Identification and Management of Patients with Suspected Sepsis in UK NHS Hospitals: A Survey of Healthcare Professionals

**DOI:** 10.3390/antibiotics9110737

**Published:** 2020-10-26

**Authors:** Amanda Winter, William Stephen Jones, A. Joy Allen, D. Ashley Price, Anthony Rostron, Raffaele Filieri, Sara Graziadio

**Affiliations:** 1NIHR Newcastle In Vitro Diagnostics Co-Operative, The Medical School, Newcastle University, Framlington Place, Newcastle upon Tyne NE2 4HH, UK; joy.allen@newcastle.ac.uk (A.J.A.); sara.graziadio@newcastle.ac.uk (S.G.); 2The Newcastle Hospitals NHS Foundation Trust, Royal Victoria Infirmary, Queen Victoria Road, Newcastle Upon Tyne NE1 4LP, UK; david.price15@nhs.net; 3Translational and Clinical Research Institute, Newcastle University, Newcastle upon Tyne NE2 4HH, UK; will.jones@newcastle.ac.uk (W.S.J.); anthony.rostron@newcastle.ac.uk (A.R.); 4Integrated Critical Care Unit, Sunderland Royal Hospital, South Tyneside and Sunderland NHS Foundation Trust, Kayll Road, Sunderland SR4 7TP, UK; 5Audencia Business School, Marketing Department, 8 Route de la Jonelière, B.P. 31222, 44312 Nantes, CEDEX 3, France; raffaele.filieri@audencia.com

**Keywords:** survey, development, diagnostic, care pathway, antibiotic stewardship, sepsis, near-patient test

## Abstract

Development of a new diagnostic is ideally driven by an understanding of the clinical need that the test addresses and the optimal role the test will have within a care pathway. This survey aimed to understand the clinical need for new sepsis diagnostics and to identify specific clinical scenarios that could be improved by testing. An electronic, cross-sectional survey was circulated to UK National Health Service (NHS) doctors and nurses who care for patients with suspected sepsis in hospitals. Two hundred and sixty-five participants completed the survey, representing 64 NHS Trusts in England. Sixty-seven percent of respondents suggested that the major cause of delay was during the initial identification of sepsis and the subsequent recognition of patients who were deteriorating. Existing blood tests did not enhance the confidence of consultants making their diagnoses. Those surveyed identified a role for a near-patient test to “rule out” suspected sepsis and, thereby, stop or postpone use of antibiotics. Current diagnostic tests are slow, non-specific, and do not reliably identify patients with a high suspicion of sepsis. As a result, they have a limited use in patient management and antibiotic stewardship. Future development of sepsis diagnostics should focus on overcoming these limitations.

## 1. Introduction

The Third International Consensus Definitions for Sepsis and Septic Shock define sepsis as “life-threatening organ dysfunction caused by a dysregulated host response to infection” [1]. Sepsis is widely cited as one of the biggest challenges faced in healthcare: it is estimated that the mortality rate of sepsis in the UK is 29% [2], with approximately 46,000 people dying of sepsis annually. The cost to the UK National Health Service (NHS) is estimated to be £1.1 billion per year [3]. As a result of these figures, it is unsurprising that many companies, and academics alike, are interested in developing in vitro diagnostics for sepsis, even though this is challenging, as sepsis is a syndrome, not a disease.

To develop diagnostics that are fit for purpose, it is important to have a comprehensive understanding of the current care pathway (or diagnostic pathway) for the disease in question. Understanding the current diagnostic pathway can help to identify any current unmet clinical needs, which can drive innovation. Articulating an unmet clinical need will identify the optimal patient population and the role for a new diagnostic, i.e., as triage, add-on, or replacement test; this information is crucial for the technical development of the new diagnostic and in planning for further evidence generation, attaining regulatory approvals, and facilitating adoption in the healthcare system of interest [4,5,6]. Establishing the current pathway is a complicated, multi-faceted process, and includes undertaking systematic reviews of published literature and guidelines, audits, expert interviews, and surveys [6]. The last of these, surveys, can help to gather attitudes, opinions, and elicit descriptive data from large cohorts of experts, and, through online dissemination, can be easily administered across wide geographical areas quickly [7].

The survey described in this article was designed to validate a draft care pathway, which was developed following UK national sepsis guidelines (National Institute for Clinical Excellence, NG51) [8], and the Sepsis Six Pathway [9]. Additionally, the survey sought to establish the situations where there is the greatest need for a test by eliciting opinions from healthcare professionals with experience in identification and management of patients with suspected sepsis. The results of this helped to pinpoint the characteristics, which a diagnostic test would need to address unmet clinical needs. It is hoped this will motivate and stimulate innovation in new sepsis diagnostic test development, and aid UK NHS Trust managers in identifying solutions to improve the efficiency of care pathways.

## 2. Results

### 2.1. Demographic Data

The survey was completed by 265 health and social care professionals, across 64 distinct UK NHS Trusts. The majority of the respondents were doctors (*n* = 209) with most declaring intensive care or hospital wards as their place of work. Anaesthesia (*n* = 38) and intensive care (*n* = 65) were the most common primary specializations. The majority of respondents had seen over 50 cases of suspected sepsis (*n* = 188) and had more than 10 years of post-qualification experience (*n* = 162). For full demographic data, see Table 1.

Definitions of sepsis, suspected sepsis, the National Early Warning Score (NEWS), and red flag signs and symptoms are given on page 2 of the Appendix A. Participants were requested to refer to these definitions when answering the survey questions.

### 2.2. Clinical Need

Seven different scenarios were proposed where a test could be used within the care pathway for diagnosis and management of sepsis. These covered potential roles as a prognostic or predictive test as well as a diagnostic test: Test for pre-hospital referral;Diagnostic test to rule in infection in the hospital to inform patient treatment;Diagnostic test to rule out infection in the hospital to inform patient treatment;Prognostic test in the hospital to escalate care;Predictive test to de-escalate antibiotics;Test to identify patients who can be discharged from intensive care unit (ICU);Test for discharging people from hospital.

The survey respondents were asked to assign a score between 1 and 10 on the clinical need for a test in each of these scenarios (1 where the test would have least use and 10 the most use).

The mean scores for clinical need, split by job role and section in the hospital where the respondents worked, are shown in Figure 1. The highest mean score was 8.22 (SE ± 0.12), which was for a test that would rule out infection in patients with NEWS ≥ 5, with results informing the decision to stop or postpone antibiotics. This was closely followed by a rule-in test with a score of 8.05 (SE ± 0.11) for patients with NEWS ≥ 5, which would inform a decision on whether to start or continue antibiotics. The least popular applications were informing discharge from ICU and from the hospital. Nursing staff gave higher average scores for the clinical need scenarios, (sample mean = 8.2), and consultants were most cautious in their average estimates of value, (sample mean = 7.1).The scenario where a test would be most useful in the opinion of doctors (consultant and trainee) was as a rule-out test. Trainee doctors gave a score of 7.91 (SE ± 0.22) and consultants a score of 8.40 (SE ± 0.16). Nursing staff preferred a rule-in test, assigning a score of 8.67 (SE ± 0.20). ICU staff gave the highest scores for clinical need across three of the seven scenarios: rule-out 8.47 (SE ± 0.18), rule-in 8.3 (SE ± 0.17), and de-escalation of antibiotics 8.29 (SE ± 0.15).

One of the aims of the survey was the identification of the characteristics that a test might need to be useful in clinical practice. An important aspect for clinical tests is whether there is an advantage for healthcare professionals for a near-patient test versus a laboratory test. In the health and social care professionals surveyed, there was a strong preference for near-patient testing devices to facilitate the identification of sepsis, as opposed to laboratory tests: 77% (202/263) preferred a near-patient test, 12% (32/263) a lab test, and 11% (29/263) were not sure. When asked to state a reason for this selection, the majority of those who preferred a near-patient test stated rapidity of result. Indeed, 87% (230/263) said a test that detects infection and predicts deterioration within 10 min would be useful, and only 2% (5/263) said it would not be useful. Reasons for stating it would not be useful included: concerns about accuracy (sensitivity and specificity); 10 min is too slow (2 min preferred) and preferring to rely on clinical judgment. Of 12% (32/263) who preferred a laboratory-based test, the reasons given were that they are more accurate and reliable and had better quality control. Some of the reasons participants gave for being unsure about rapid [11% (28/263)], or near patient testing [11% (29/263)], was that they felt there was a trade-off to manage between turnaround time and accuracy/reliability when using these tests. Other explanations respondents gave were that the clinical picture was still more important and that additional testing could cause dependence and additional delay.

### 2.3. Care Pathway

Once a patient has been identified as having suspected sepsis at the bedside, by signs and symptoms and early warning scores, the next steps for that patient generally include blood tests to assess levels of specific biomarkers, and collection of samples for culture. The survey also sought to investigate aspects such as time to response and impact on clinical decision making.

Data was gathered on how patients with suspected sepsis were flagged by nurses to attending doctors (For glossary of definitions and acronyms, see Appendix A). Approximately 64% (90/140) of nurses and trainee doctors stated that their pathway flagged patients only for deterioration but that this was not specific for sepsis. A further 26% (36/140) stated that they had a pathway that specifically alerted for sepsis, and the remainder were unsure. Of the responses, 28% (66/239) indicated that patients with suspected sepsis are most often identified using the NEWS score alone; 21% (49/239) used NEWS score in combination with clinical suspicion of infection; 15% (35/239) used red flags alone; 13% (31/239) used NEWS score or a red flag and suspicion of infection; and 13% (30/239) used red flag and suspicion of infection. Around 11% (25/239) stated they were unsure of their current practice. 

Consultants were asked when they would start antibiotics on a patient with NEWS ≥ 5 or with a red flag present. The majority (97% (110/114)) reported that they would start antibiotics when the patient shows signs or symptoms consistent with infection, and the remaining 3% (4/114) reported they would start only when the results of blood tests were consistent with infection. Trainee doctors were asked how often blood cultures are taken before antibiotics were administered. Around 18% (15/84) said that blood cultures were always taken first and a further 72% (60/84) said this usually happens. Only 6% (5/84) said blood cultures were taken first sometimes or rarely and 5% (4/84) were unsure.

A set of questions which addressed the amount of time taken to complete key tasks across the care pathway were included, the results of which are summarised in Table 2. Approximately 14% (18/135) of respondents reported that it takes longer than one hour to administer antibiotics, with 83% (29/35) of respondents reporting that antibiotics were administered within one hour if they had a review procedure specifically for suspected sepsis. However, this fell to 60% (53/87) of respondents when there was a procedure for deterioration only.

All respondents were asked where they thought the major sources of delay were when managing patients with suspected sepsis, and what the underlying reasons were for these setbacks. These are summarised in Figure 2.

The majority, 67% (171/257), felt that the main source of delay was the difficulty in identifying patients with suspected sepsis. The remaining 40% (102/257) said that flagging up of deteriorating sepsis patients was the main source. When asked what the main reason for this was, 50% (124/246) respondents cited the shortage of nurses and the remaining 50% (122/246) respondents suggested the lack of junior doctors. Another factor, which was selected by 34% (84/246) of the participants was the lack of rapid diagnostic tests: around 43% (23/53) of nurses said that this was a problem. Less than 15% (34/257) of respondents believed that there were no major delays, whilst 24% (13/54) of nurses, 22% (19/86) of trainees, and only 14% (16/117) of consultants thought there were delays only out of hours. Consultants were asked how the decision to stop antibiotics would be made. From those surveyed, 42% (46/110) said that antibiotics would be stopped when the patient improves clinically *and* this improvement was consistent with test results. However, 30% (33/110) of consultants said that they would complete the course of antibiotics alongside clinical improvement of the patient and blood results. Irrespective of blood test results, 16% of consultants (17/114) would stop when the patient has completed the course of antibiotics and 11% (12/110) would stop when the patient improved clinically.

The percentage of patients with sepsis who die despite appropriate and timely management was estimated by the consultants. Around 70% (77/111) of consultants said that this happens in <25% of cases.

### 2.4. The Use of Laboratory Tests

The use of lactate, C-Reactive Protein (CRP), and procalcitonin (PCT) blood tests for the diagnosis and monitoring of patients with suspected sepsis were explored in the survey.

#### 2.4.1. Diagnosis

Lactate is a popular method for assisting diagnosis, with 71% (60/84) of respondents saying this is performed on more than 75% of their patients with suspected sepsis. Around 80% (92/116) of consultants estimated that 75% of patients have a CRP test performed to support a diagnosis of sepsis. PCT tests was reported to be less commonly used, with 57% (65/115) saying they never used this method for diagnosis. Other tests that were used in the diagnosis of sepsis were as follows with number of respondents citing these tests in brackets: full blood counts (including white cell counts, white cell differential and platelet count, (82/122)), cultures (including blood culture, (20/122)); urea & electrolytes (12/122); liver function tests (8/122); arterial blood gases (8/122) and coagulation tests (8/122). The perceived helpfulness of existing tests in making a diagnosis was assessed: 19% (22/114) felt the tests were ‘very helpful’, 78% (89/114) ‘somewhat helpful’ and 3% (3/114) ‘not very helpful’ at all.

#### 2.4.2. Monitoring

The respondents reported that CRP was used less to support the decision to postpone antibiotics than to support a diagnosis of suspected sepsis, with 57% (67/117) of respondents using it to postpone the use of antibiotics in less than 25% of cases. There was no consensus over the use of CRP to inform the decision to stop antibiotics, with 27% (32/117) never using CRP to guide this decision, 34% (40/117) using it on less than 50% of patients, and another 33% (38/117) using it on more than 50% of their cases. Around 70% of respondents reported that they have never used PCT, or they used it in less than 25% cases, when deciding whether to postpone or stop antibiotic treatment. The additional tests that were used for monitoring were similar to those selected for diagnosis (with the number of respondents citing these tests in brackets): full blood counts (54/122), lactate (10/122), urea and electrolytes (10/122), cultures (7/122), and liver function tests (6/122). The majority felt these tests were ‘somewhat helpful’ in monitoring patients (83%, 90/109).

#### 2.4.3. Test Availability

According to the respondents who were trainee doctors, 92% (76/83) of NHS Trusts have an onsite laboratory for both haematology and biochemistry. Lactate results were stated to be available within 5 min for 22% (18/82) of participants and within 1 h for 66% (54/83). Lactate results took greater than 1 h for 24% (20/83) of trainee doctor respondents while 10% (8/83) of the respondents were unsure about turnaround time. For haematology and biochemistry results, 45% (38/84) had results returned in around 1 h or less, and 43% (36/84) between 1 and 2 h. Only 6% (5/84) of respondents reported results taking longer than 2 h. Around 48% (40/84) of respondents reported that blood culture results were typically available after 48–72 h. Around 36% (30/84) of respondents were able to obtain results within 24–48 h, and 4% (3/84) within 24 h. The same proportion [4% (3/84) of respondents] received results after 72 h.

Figure 3 shows how the number of cases, which can be confidently identified as sepsis increases as further information is made available to the decision maker. Most consultants reported that, when using clinical judgement alone, 51–75% of sepsis cases could be confidently identified. This rose to 75–100% when the results of blood tests and cultures were made available. Only 17% of respondents reported that all cases could be confidently identified even retrospectively.

## 3. Discussion

The results of this survey demonstrate that there are unmet clinical needs to drive innovation in new diagnostics for sepsis. The greatest needs for tests are for: assisting in the identification of new cases, monitoring deterioration, and accelerating the escalation of patients from initial flagging up of deterioration by nursing staff to the eventual administration of antibiotics. According to the results of this survey, this could be best achieved by a near-patient test to improve turnaround time. A result within ten minutes was considered helpful by the majority of respondents.

The National Institute for Clinical Excellence (NICE) currently recommends that antibiotics are administered to patients who have high-risk signs and symptoms within one hour of meeting a high-risk criterion in secondary care [8]. In order to achieve this, a simplified care bundle pathway, Sepsis 6, is used in many trusts across the UK. This has been associated with improved mortality rates in patients with suspected sepsis [9]. Prompt first medical and senior doctor review is correlated with increased compliance with the Sepsis 6 pathway [10]. However, the majority of health and social care professionals surveyed here recognised that the tools which they currently have to identify sepsis and deterioration were not adequate and caused delays to the management of sepsis patients. The lack of rapid diagnostics for sepsis was stated as the third most common underlying cause of these delays. This was most strongly articulated by nurses, who might feel the greatest burden of initial flagging up of sepsis. It is also important to note that most respondents pointed out that the two main causes of delay are shortages of nursing staff and junior doctors. Without addressing the shortage of the personnel, the impact of new diagnostics may therefore be limited. Diagnostic tests are rarely used in isolation, and full downstream patient benefits of such tests can only be realised when they are fully integrated into care pathways as complex interventions.

The majority of those surveyed indicated that they were flagging up suspected sepsis patients, reviewing them and administering tests and treatment in a timely manner, as per the NICE guidance. One of the interesting findings of this survey was a difference between NHS trusts where their policy was to flag up patients for deterioration only vs. specifically suspicion of sepsis. This appeared to be associated with decreased time to antibiotic administration. In a prospective study examining performance of diagnostic and screening tools for sepsis, the choice of screening tool used (Sequential Organ Failure Assessment, Red Flag Sepsis, etc.) did not make any difference to the completion of the Sepsis Six, and the use of *any* screening tool improved survival [11]. Diagnosis of sepsis applies to identifying it both in newly sick patients and in those who have decompensated existing infections, particularly since all early warning scores rely on physiological changes to trigger review. The ability of nursing staff to recognise septic patients earlier may increase chances of survival [12]. Therefore, point-of-care tests which can be carried out by nurses to identify those patients who need medical escalation (i.e. a rule-in test) may be beneficial.

It is difficult to objectively measure how frequently sepsis is accurately diagnosed at the bedside, even with the benefit of retrospective analysis. The consultants’ perception of what proportion of sepsis patients could be correctly identified increased when the results of initial blood tests were known. However, there is, perceived to be, little incremental value gained from the addition of culture results (only a decrease in the variability) or even the benefit of knowing the final patient outcome. This indicates that much of the diagnostic information may come from clinical assessment and blood tests, but the uncertainty of the diagnosis is still very high in acute care. This is supported by clinicians in this survey, the majority of whom pointed out that blood tests, such as CRP and lactate, were commonly performed but were only ‘somewhat helpful’, and that commencement and cessation of antibiotics is mostly based on clinical assessment. The recommended durations of antibiotic therapy for sepsis has little evidence rationale; indeed, CRP- and PCT- guided reduction in antibiotic therapy duration is currently being investigated in the ADAPT-Sepsis trial and is due to complete in April 2021 [13].

The target for rapid administration of antibiotics driven by guidelines, such as those from NICE, is often in conflict with the need for antibiotic stewardship, and tests that can help resolve this are required. There is much debate amongst the sepsis community over the introduction of time targets to administer antibiotics, with some stating that they have reduced preventable deaths and others arguing that they drive the overuse of broad-spectrum antibiotics [14]. The axis model proposed by Prescott and Iwashyna suggests a framework for the timing and broadness of antibiotic therapy. Patients who are sickest should receive antibiotics as soon as possible, but if there is diagnostic uncertainty about whether infection is the cause, then delays are possible. Conversely, patients for whom there is greater diagnostic certainty but are less sick may end up with more prompt treatment. The setting in which the intervention is being made is also likely to be important. Those in high-intensity care settings where ‘watchful waiting’ is less risky (because of high-frequency observations and resource/staff availability for quick interventions on the basis of these observations) may be more inclined to do so than those in settings where nursing and observation capacity is lower. There is unlikely to be a right time to administer antibiotics which will fit all patients, and we currently lack the tools to have certainty very quickly about many sepsis diagnoses at presentation. As this survey has demonstrated, the ability of the clinician to have absolute diagnostic certainty is unlikely even in retrospect. However, it is nevertheless beneficial to apply retrospective scrutiny over decision making and treatment to help to maintain good antibiotic stewardship.

A diagnostic was preferred as a ‘rule out’ test by medics, which may reflect both the difficulty of excluding sepsis as a possible differential diagnosis in patients who have non-specific symptoms and also the potential serious consequence of missing a case of sepsis. The consequences of a missed diagnosis are often immediately apparent to the prescriber e.g. death, or acute organ failure. However, the longer term ‘risks’ associated with unnecessary treatment of a false positive, e.g. side effects, treatment complications, diagnostic blindness and antimicrobial resistance are not always obvious, and they do not necessarily influence the outcome of the patient, so this may lead to the overestimation of treatment benefit [15]. The sensitivity of existing biomarkers CRP and PCT (for diagnosis of sepsis) are around 75% (95% CI 69-79) and 79% (95% CI 75-83) respectively [16], but the fact that doctors still wanted a rule-out test (despite CRP being widely used), suggests that those figures are not high enough to satisfy that need. It has been stated that biomarkers for sepsis should be aiming for Area Under the Receiver Operating Characteristic(AUROC) values of 0.9 or higher [17]. The possible benefits of an accurate rule-out test are: reduction in prescription of antibiotics, decreasing use of imaging procedures to search for the source and to promote consideration of alternative diagnoses [18]. Existing biomarkers do not generally possess high enough specificity to differentiate infection from other inflammation aetiologies [16,19]. This limits their usefulness in practice. There is also a gap in basic research to identify new biomarker(s), which possess the required characteristics.

The majority of respondents were able to access laboratory results within typical timescales (<60 min) [20], but these are currently too slow to be of any utility for reaching the target of administering antibiotics within one hour. There was a clear preference amongst those surveyed for near-patient testing vs. standard laboratory tests because they felt that near-patient tests would be faster than a laboratory test. C. Price noted that the benefit of near-patient devices is that there is temporal proximity of performing the test and receiving the result i.e. rapid turnaround, which subsequently may inform quicker action [21]. In this scenario, there is less opportunity for distraction with other tasks whilst there are gaps between stages. Point-of-care devices which provide sensitivity and specificity >90% within one hour are not available [22] and there is a need to have information from diagnostic tests quickly, *before or during* the process of deciding to treat/escalate a patient.

### Limitations

It has not been possible to calculate a response rate, despite the efforts of the authors, due to the nature of the dissemination routes and sampling. Any estimate would have been misleading and, therefore, it was deemed more robust not to report the denominator. The scores for almost every clinical scenario the participants could select were above seven, indicating that there is not just one stand-out issue in sepsis management and the problem is somewhat unfocused. The amount of uncertainty arising in the answers to a number of questions was quite high, which may be because the information sought in the survey is not widely known amongst clinicians, that it isn’t monitored or reported in their trust, or that it is difficult for those reporting to make accurate estimates. This only affected the sample size in some answers, however, not their accuracy. We reported only the answers where enough respondents were confident enough to choose an option. Most staff are familiar with the NICE guideline and the targets set within it, which may have caused biased answering. However, the target to administer antibiotics within one hour remains controversial amongst many senior sepsis experts [23,24], so the bias might be mitigated. Severe acute respiratory syndrome coronavirus 2 (SARS-COV-2) was not circulating when the survey was carried out, so the perception of clinicians may have changed. However, since the symptoms of SARS-COV-2and sepsis can overlap in some patients, it seems plausible that the need for rapid and accurate diagnostics for sepsis is even more important in the current situation.

## 4. Materials and Methods

The project received Heath Research Authority approval (IRAS number: 230491), and was adopted by the National Institute for Health Research (NIHR) Clinical Research (CRN) portfolio (CPMS 35683).The survey was developed in collaboration with a clinical expert in the topic area (AR), and was tested on a pilot sample of 10 members of the target group of respondents. Feedback was considered and the questions were modified as appropriate.

Any UK registered doctor or nurse who had experience in identifying and/or managing patients with suspected sepsis were eligible to take part in the survey. An introductory cover letter was included at the start of the survey, specifying details about the organization behind the study, contact details and name of researcher, ethical approvals, aims, inclusion criteria, and consent mechanism. Definitions of sepsis, suspected sepsis, the National Early Warning Score and red flag signs and symptoms were provided at the beginning of the survey. The survey collected demographic data including the respondent’s job role, professional experience and geographical location, in addition to the main study questions on their clinical practice in identification and management of patients with sepsis. These included aspects such as current care pathways in sepsis, current availability and utility of tests for sepsis, and their opinion on the current unmet clinical needs in this pathway (the full survey is given in Appendix A). Respondents to the survey were asked specific questions based on their job role. Therefore, nurses, trainee or career grade doctors, and consultants all answered different combinations of questions. Conversely, all participants answered questions about potential clinical need for novel tests, placement of the test in the pathway, and causes of delay in the current care pathways for diagnosis and management of sepsis. Questions that related to practical aspects of the recognition and management of sepsis patients were directed to nursing staff and junior medical staff (trainee or career doctors; in the result section we refer to this category as “trainee”). Questions that related to the higher-level decision making and conceptualization behind diagnosis, management and outcomes were directed at consultants. Participants could skip questions if they wished.

In some questions, respondents were asked to estimate frequencies. They were given a choice of six categories (0%, 1–25%, 25–50%, 50–75%, 75–99%, 100%) and “unsure”. The frequency bands broadly match to “never”, “rarely”, “sometimes”, “often”, “usually”, “always”.

The electronic survey was distributed by the NIHR CRN, and by networks of collaborators in the project, obtaining a convenience sample. Any potential respondent meeting the inclusion criteria was invited to participate. The survey was available to complete between August 2017 and December 2017. Both quantitative and qualitative data were collected for analysis. Descriptive statistics are presented in the form of percentages and means (with standard deviation).

## 5. Conclusions

There are unmet needs within the care pathways for diagnosis and management of sepsis in the UK NHS. This survey has identified problems faced by health and social care professionals in the diagnosis and management of sepsis which should be used to inform research and development of novel sepsis diagnostic tests. Key findings include the preference of trainee doctors and consultants for testing to ‘rule out’ sepsis, in contrast to nursing staff who favoured testing to ‘rule in’ sepsis. The results of this survey also highlighted that currently available biomarkers have limited utility when used to inform difficult clinical decisions such as starting or stopping antibiotics. Having a rule-out test could support diagnosis, and appropriate antibiotic stewardship. These findings could be a useful starting point from which to develop tests which address the unmet needs of the UK NHS, supporting improved patient care through more informed clinical decision making. 

## Figures and Tables

**Figure 1 antibiotics-09-00737-f001:**
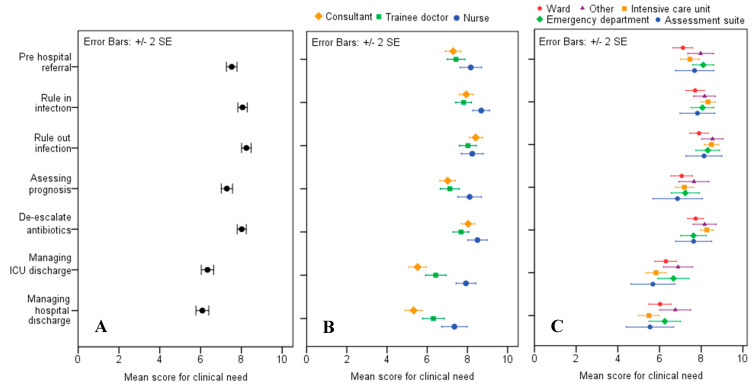
Scores of clinical need for a novel test in seven different clinical scenarios. (**A**) Overall mean scores for clinical need for seven different clinical scenarios. (**B**) Mean scores for clinical need for categorised by job role. (**C**) Mean scores for clinical need categorised by hospital section.

**Figure 2 antibiotics-09-00737-f002:**
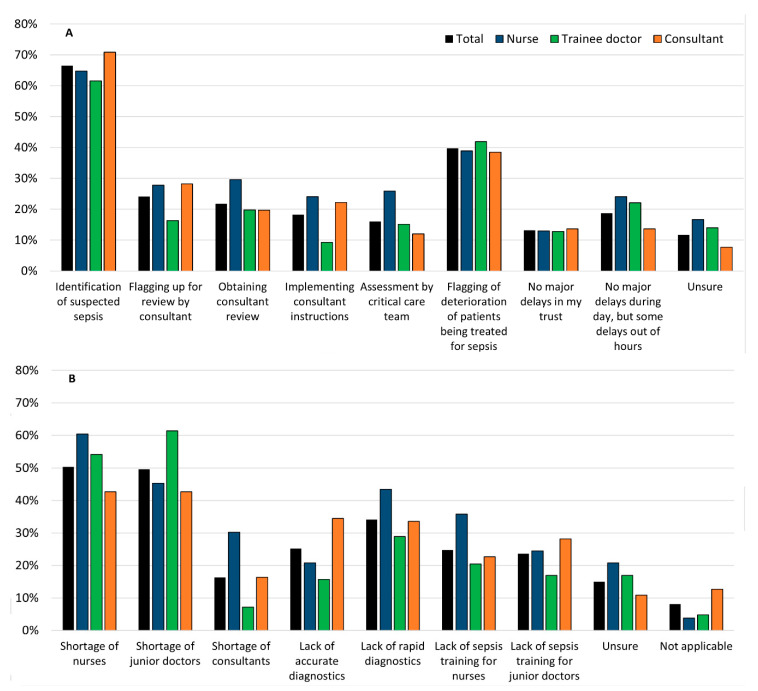
Percentage of respondents identifying (**A**) common delays in management of sepsis patients and (**B**) causes of delays.

**Figure 3 antibiotics-09-00737-f003:**
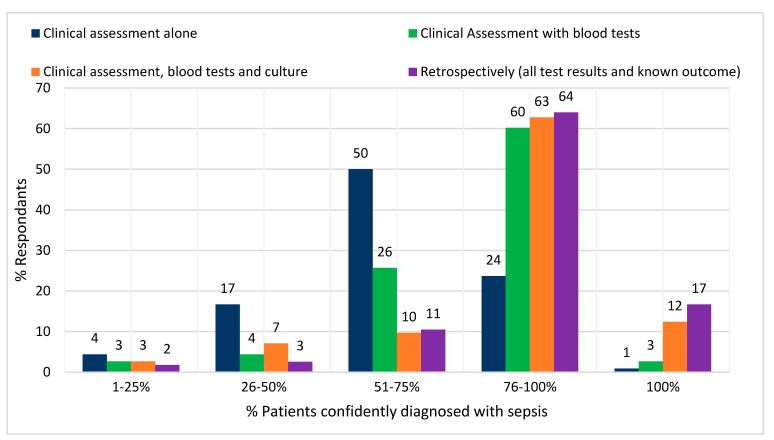
Percentage of consultants estimating what percentage of sepsis patients could be confidently identified using temporal sequence with increasing information.

**Table 1 antibiotics-09-00737-t001:** Characteristics of respondents to online survey.

**Total Number of Respondents**				265
**Gender**				
**Male (%)**	51.3	Female (%)		39.2
Preferred not to say/unanswered (%)		9.5		
Job Role				
Nurse				56
Trainee or career grade doctor				87
Consultant				122
Patient age group				
Adults	235	Paediatrics		29
Hospital sections covered				
Admissions unit	23	Ward		87
Emergency department	53	Others		37
Intensive care unit	92			
Years of post-qualification experience				
Less than 1	15	Between 1 and 4		49
Between 5 and 9	37	10 or more		162
Primary Specialisation				
Anaesthesia	38	Oncology		9
Infectious diseases	2	Surgery		15
Microbiology	6	Paediatrics/Neonatology		21
Acute medicine	22	None, e.g., trainee on rotation		33
Intensive care	65	Emergency Medicine		20
Surgery	16	Geriatrics		7
Obstetrics/gynaecology	4	Haematology		4
General Practise	2	Cardiology		2
Gastroenterology	2	Other		8
Type of hospital				
Tertiary or teaching hospital	141	Tertiary/Teaching *and* District General Hospital		15
District General Hospital	100	Other/unknown		15
Number of unique NHS trusts (all in England)				64
Number of suspected sepsis cases seen				
Less than 5	7	Between 5 and 50		69
More than 50	188	Unsure		1

**Table 2 antibiotics-09-00737-t002:** Estimation of time taken to complete key tasks in the care pathway of patients with sepsis. The option that has been selected more often by the respondents is highlighted in bold.

Key Tasks in Care Pathway	Amount of Time Taken to Complete (Min)				
	0–15	15–30	30–60	>60	Unsure
Time from initial flagging to first review by a clinician who can take decisions on the next step of management.	11% (15/140)	**32% (43/140)**	26% (39/140)	4% (5/140)	27% (38/140)
Time from initial flagging to blood for further analysis and culture being taken.	16% (13/84)	**33% (28/84)**	27% (23/84)	6% (5/84)	18% (15/84)
From the moment the patient is flagged by the nurse for review, how long does it usually take before the patient is given antibiotics?	1% (2/135)	16% (20/135)	**47% (63/135)**	14% (18/135)	22% (32/135)
Time taken from request to assessment for potential ICU admission (if requested).	3% (4/138)	20% (27/138)	**30% (41/138)**	6% (9/138)	41% (57/138)

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
