# Peer review of "The Clinical Need for New Diagnostics in the Identification and Management of Patients with Suspected Sepsis in UK NHS Hospitals: A Survey of Healthcare Professionals"

_antibiotics, 2020, doi:10.3390/antibiotics9110737_

Round 1

Reviewer 1 Report

Winter et al outline the results of an online survey of UK health professionals around current practices and perceived needs in diagnostics. Overall, the manuscript is well-written, timely, and I have no major concerns. Sincere congratulations to the authors on this. I do have some minor concerns that are worth addressing, as follows:

  1. How many people were asked to participate in the survey, and what does that mean about the self-selected group that chose to participate (what percentage participated)? Are these results ‘too rosy’ because of the selected group?
  2. Could the authors opine a bit on whether the diagnosis of sepsis is primarily about new infections in sick patients, new decompensation in patients with infections, or both? And what is the right time to administer antibiotics? Sepsis has two ‘axes’ of severity and infections, and I would point the authors to another good work in this field by Hallie Prescott, PMID 30883190.
  3. What was the role of the industry sponsor in the design of the survey?
  4. Lactate appears to be listed twice with different numbers, see lines 171 & 176.

Author Response

Winter et al outline the results of an online survey of UK health professionals around current practices and perceived needs in diagnostics. Overall, the manuscript is well-written, timely, and I have no major concerns. Sincere congratulations to the authors on this.

Thank you.

Suggestions are addressed as follows:

1. How many people were asked to participate in the survey, and what does that mean about the self-selected group that chose to participate (what percentage participated)? Are these results ‘too rosy’ because of the selected group?

The way that the survey was disseminated means it is impossible for us to provide an estimate of the number of participants invited to complete the survey. This has been added to the study limitations (Line 333-335).

2.Could the authors opine a bit on whether the diagnosis of sepsis is primarily about new infections in sick patients, new decompensation in patients with infections, or both? And what is the right time to administer antibiotics? Sepsis has two ‘axes’ of severity and infections, and I would point the authors to another good work in this field by Hallie Prescott, PMID 30883190.

Thank you for the recommendation for further reading. We have made some adaptations to the discussion on the basis of your questions raised and have included the recommended paper.

    • Diagnosis of new infections vs decompensation in existing cases: Lines 268-270
    • The right time to administer antibiotics: Lines 287-304

3. What was the role of the industry sponsor in the design of the survey?

Please see conflict of interest statement. This statement has now been strengthened in the manuscript (line 403).

4. Lactate appears to be listed twice with different numbers, see lines 171 & 176.

Thank you for pointing this out, there was an error in the survey logic which has now been removed (line 201).

Reviewer 2 Report

Thank you for the oppertunity to review this paper. The overall presentation is good and clear. It should however be noted that the describe results have merely interest to NHS, and limited interest to other healtcare systems, even in the EU due to the substantial differences in organization and ressources. This must be taken into consideration when deciding upon publication. Further, the current biomedical understanding of sepsis clearly lack good rule in / out tests for sepsis - and it is thus of limited interest and very hypothetical what the clinicans would like.

Please provide the invited / responded rates for the different clincian groups

Author Response

Thank you for your comments and suggestions.

1. The overall presentation is good and clear.

Thank you. This does seem to be at odds with the selection of 'extensive editing of English language and style required'. In response ee have had the spelling and style checked by an experienced member of our team who was not involved with the original research.

2. It should however be noted that the describe results have merely interest to NHS, and limited interest to other healtcare systems, even in the EU due to the substantial differences in organization and ressources. This must be taken into consideration when deciding upon publication.

The authors are supported by the National Institute for Health Research and as such our research projects centre largely around the UK NHS. We don’t believe that the findings of this research are limited to this healthcare system, as there are significant overlaps in many aspects of sepsis identification and management, as good evidence-based medicine dictates. (Dizon, J.M., Machingaidze, S. & Grimmer, K. To adopt, to adapt, or to contextualise? The big question in clinical practice guideline development. BMC Res Notes 9, 442 (2016). NICE guidelines are well received throughout the world, indeed, they offer a service which allows them to be licenced in other countries outside of the UK (https://www.nice.org.uk/about/what-we-do/international-services/using-our-content-outside-the-uk) The current work sets out unmet research priorities (such as basic biomarker research, the need of which is not a problem exclusive to the UK. The international community can be part of the solution, if a problem is identified. The NHS, whilst in many aspects is a unique healthcare system, we believe that access to a single healthcare system with large population of interest is an ideal setting to do the research.

3. Further, the current biomedical understanding of sepsis clearly lack good rule in / out tests for sepsis - and it is thus of limited interest and very hypothetical what the clinicans would like.

The authors agree wholeheartedly with the assessment the current biomedical understanding of sepsis is incomplete. Indeed, a paragraph in our discussion (lines 305-321) is dedicated to this point.

4. Please provide the invited / responded rates for the different clincian groups

The way that the survey was disseminated means it is impossible for us to provide an estimate of the number of participants invited to complete the survey. This has been added to the study limitations (Line 333-335).

Reviewer 3 Report

The paper is well written and interesting. I cannot beleive that there is a low knowledge and usage of PCT for diagnosing/monitoring sepsis and stopping antibiotics... Anyway these are the results...

My greater concern is about the appropriateness of the paper with the topic of the present Journal.

Author Response

Thank you for the positive review.

1. The paper is well written and interesting.

Thank you.

2. I cannot beleive that there is a low knowledge and usage of PCT for diagnosing/monitoring sepsis and stopping antibiotics... Anyway these are the results...

The authors have rechecked the responses in relation to PCT and they are correct. Some minor changes have been made to the text to clarify that we are talking about PCT (Line 210-212)

3. My greater concern is about the appropriateness of the paper with the topic of the present Journal.

As sepsis is a key condition which drives the administration of broad-spectrum antibiotics, we hope that disseminating the findings of this research amongst Antibiotics’ readership will help to identify areas where improvements can be made. The link to antimicrobial prescription and sepsis has been strengthened in the discussion (lines 287-304).

Reviewer 4 Report

In this study, Winter and colleagues carried out a survey in UK NHS hospitals to understand the clinical need and potential roles for new diagnostic tests for sepsis, and also to identify clinical scenarios that could be improved by diagnostic testing. They found that current diagnostic tests are slow, non-specific and do not reliably identify patients with a high suspicion of sepsis. The results may help support test developers in their R&D process, and UK Trust managers in identifying solutions to improve the efficiency of care pathways.

The study was well presented and results were interpreted properly with limitations discussed thoroughly. 

Author Response

1. In this study, Winter and colleagues carried out a survey in UK NHS hospitals to understand the clinical need and potential roles for new diagnostic tests for sepsis, and also to identify clinical scenarios that could be improved by diagnostic testing. They found that current diagnostic tests are slow, non-specific and do not reliably identify patients with a high suspicion of sepsis. The results may help support test developers in their R&D process, and UK Trust managers in identifying solutions to improve the efficiency of care pathways.

The study was well presented and results were interpreted properly with limitations discussed thoroughly.

Thank you for the positive review.